# Structural Analysis of the Simultaneous Activation and Inhibition of γ-Secretase Activity in the Development of Drugs for Alzheimer’s Disease

**DOI:** 10.3390/pharmaceutics13040514

**Published:** 2021-04-08

**Authors:** Željko M. Svedružić, Katarina Vrbnjak, Manuel Martinović, Vedran Miletić

**Affiliations:** 1Department of Biotechnology, University of Rijeka, 51000 Rijeka, Croatia; katarina.vrbnjak@kuleuven.be (K.V.); manuel.martinovic69@gmail.com (M.M.); 2Laboratory for Medical Biochemistry, Psychiatric Hospital Rab, Kampor 224, 51280 Rab, Croatia; 3Laboratory for Mechanisms of Cell Transformation (VIB-KU Leuven), ON IV Herestraat—Box 912, 3000 Leuven, Belgium; 4Department of Informatics, University of Rijeka, 51000 Rijeka, Croatia; vmiletic@inf.uniri.hr

**Keywords:** Alzheimer, semagacestat, avagacestat, molecular dynamics, drug-design, familial Alzheimer’s disease, modulation of enzyme activity

## Abstract

**Significance:** The majority of the drugs which target membrane-embedded protease γ-secretase show an unusual biphasic activation–inhibition dose-response in cells, model animals, and humans. Semagacestat and avagacestat are two biphasic drugs that can facilitate cognitive decline in patients with Alzheimer’s disease. Initial mechanistic studies showed that the biphasic drugs, and pathogenic mutations, can produce the same type of changes in γ-secretase activity. **Results:** DAPT, semagacestat LY-411,575, and avagacestat are four drugs that show different binding constants, and a biphasic activation–inhibition dose-response for amyloid-β-40 products in SH-SY5 cells. Multiscale molecular dynamics studies have shown that all four drugs bind to the most mobile parts in the presenilin structure, at different ends of the 29 Å long active site tunnel. The biphasic dose-response assays are a result of the modulation of γ-secretase activity by the concurrent binding of multiple drug molecules at each end of the active site tunnel. The drugs activate γ-secretase by facilitating the opening of the active site tunnel, when the rate-limiting step is the tunnel opening, and the formation of the enzyme–substrate complex. The drugs inhibit γ-secretase as uncompetitive inhibitors by binding next to the substrate, to dynamic enzyme structures which regulate processive catalysis. The drugs can modulate the production of different amyloid-β catalytic intermediates by penetration into the active site tunnel, to different depths, with different flexibility and different binding affinity. **Conclusions:** Biphasic drugs and pathogenic mutations can affect the same dynamic protein structures that control processive catalysis. Successful drug-design strategies must incorporate transient changes in the γ-secretase structure in the development of specific modulators of its catalytic activity.

## 1. Introduction

Alzheimer’s disease is a slowly-progressing neurodegenerative disorder with a fatal outcome [1]. Alzheimer’s disease is also a major challenge for the pharmaceutical industry today, as it stands out ahead of malignant diseases as the biggest financial burden for the healthcare providers in developed countries [1,2,3].

The majority of the potential drugs for Alzheimer’s disease have targeted the metabolism of the C-terminal part of amyloid precursor protein (C99-APP) [2,4,5,6]. The target enzyme was the membrane-embedded aspartic protease: γ-secretase [2,4,5]. A large number of repeated failures in the last twenty years have led to numerous proposals that γ-secretase might not be a good therapeutic target [2,3,4,5,6]. An alternative, less frequently considered explanation is that a very few of the failed drug-design strategies have taken into account the complexity of the catalytic mechanism of γ-secretase [2,5,6,7,8].

The activation and inhibition of γ-secretase activity which can be observed in the biphasic dose-response to drugs are good examples of the complex enzymatic mechanisms that were not adequately recognized in the drug-development studies [2,7,9,10,11,12,13,14,15]. A biphasic dose-response can be observed in cell-cultures [9,11,13,16], in model animals [9,15], and in patients’ plasma in clinical trials [10]. The biphasic dose-response can be observed only in studies which use physiological sub-saturating substrate concentrations, and a full concentration range for the drugs [9,10,11,17]. It is very likely, that all types of drugs which target the presenilin subunit of γ-secretase can produce the biphasic dose-response when the Aβ metabolism is at the physiological level [7,9,10,13].

The great majority of the drug-screening and optimization studies used an unphysiological high saturation of γ-secretase with its substrate, and a limited concentration range for the drugs. Such an approach can be a good time-and money-saving strategy in the early screening process [7,9,16,18,19]. However, the ‘fast-and-cheap’ approach is also the main reason why the majority of the preclinical studies were misleading, or had poor relevance to the clinical studies [2,5,7,9,10,14,15,16]. γ-Secretase is far from saturation with its substrate in physiological conditions in cells [9,11,15,17], just as with the majority of other enzymes [18,20,21]. Sub-saturated enzymes are a fundamental physiological mechanism which can assure the fastest, best-controlled responses to metabolic changes [18,20]. Measurements that had a poorly-defined saturation of γ-secretase with its substrate can explain why so many studies have frequently reported irreproducible results on selectivity between Notch and APP substrates, or on the modulation of Aβ production [2,5,14]. Reproducible experiments depend on well-defined Kӎ values for different substrates and Aβ products [9,11,13,18,19,22,23,24].

The mechanistic studies of the biphasic dose-response drugs are also important because these compounds can produce the same type of changes in γ-secretase activity as the disease-causing Familial Alzheimer’s disease (FAD) mutations [17,25]. FAD mutation and the biphasic-drugs can cause the accumulation of the longer and more hydrophobic Aβ products [7,13,22,26,27]. At low physiological substrate concentrations, the biphasic drugs can increase the saturation of γ-secretase with its substrate, just like the Swedish mutation in its APP substrate [11]. At high substrate concentrations, the biphasic drugs act as uncompetitive inhibitors and show a decrease in the maximal turnover rates, just like the FAD mutations in presenilin 1 [11,17,22,26]. FAD mutations can affect binding constants for activation and inhibition by different drugs to different extents [17,22]. Clinical studies have shown that dose-dependent cognitive decline was observed in clinical trials with the biphasic drugs avagacestat [10,28] and semagacestat [7,16,29,30,31]. In essence, FAD mutations and biphasic drugs have the same effect on γ-secretase function: a decrease in the capacity of γ-secretase to process its substrates [17]. A decrease in the catalytic capacity of γ-secretase can produce pathogenic changes in Aβ products with the wild type γ-secretase [17,22,23,32]. A decrease in the catalytic capacity of γ-secretase appears to be shared pathogenic mechanism in sporadic and FAD cases of the disease [11,17].

Here, we present combined structure and activity studies of γ-secretase in order to describe the biphasic dose-response in the presence of the four best-known drugs. Semagacestat and avagacestat were chosen as two biphasic drugs which can facilitate cognitive decline in patients in clinical trials [7,10,14,15,16,28,31]. DAPT was chosen as a biphasic drug which has been most frequently used in mechanistic studies of γ-secretase [11,13,27,33,34,35]. LY-411,575 was chosen as a biphasic drug with some of the most potent affinities for γ-secretase [36]. We found that multiple drug molecules can bind simultaneously to the different flexible parts on the presenilin subunit of γ-secretase. The drugs bound to different sites can affect every catalytic step, from the enzyme–substrate recognition to the processive cleavages of Aβ products. We also found that the drugs and FAD mutations target the same dynamic parts in the presenilin structure.

## 2. Results

### 2.1. Biphasic Activation–Inhibition Dose-Response Curves for DAPT, Semagacestat, Ly-411,575 and Avagacestat in Cultures of SH-SY5 Cells.

We measured the biphasic dose-response in Aβ 1-40 production in SH-SY5 cells (Figure 1 [9,11,15,16]). The biphasic dose-response depends on the extent of the γ-secretase saturation with its substrate, and the assay design [9,11,17]. Thus, different drugs can be quantitatively compared only if the drugs are incubated under identical conditions in identical cells. Here, the presented measurements stand out as the measurements that have taken maximal efforts to ensure identical assay conditions for each of the four drugs (Figure 1). One batch of cells was split into four identical batches for parallel treatment with the four drugs in exactly the same conditions. The cells were incubated with each drug overnight, and the Aβ 1-40 production was measured in parallel using identical protocols.

The Aβ 1-40 production in the SH-SY5 cells treated with DAPT, semagacestat, LY-411, 575, and avagacestat show biphasic activation–inhibition dose-response curves with clear differences between the drugs in their activation and inhibition constants, and Hill's coefficients (Figure 1). Thus, the structure–activity analysis of these molecules can give insights into the mechanism behind the biphasic responses (Methods Equation (1) and [11,17]). The full numerical analysis of biphasic dose-response curves depends on six best-fit parameters (Methods Equation (1) and [11,17]). The parameters can be calculated with satisfactory accuracy if the experiments have an optimal selection of concentration ranges for each drug (Figure 1) [11,17].

We find that activation and inhibition curves always overlap; the compounds that show low EC50 values for activation also show low IC50 for inhibition (Figure 1). These results indicate concurrent binding at the activation and inhibition sites [11,12,24]. The biphasic profiles show that the Hill's coefficients can be higher than one (Figure 1), which indicates the simultaneous binding of multiple drug molecules to γ-secretase, and possible cooperativity [11,24]. The inhibition phase shows higher Hill's coefficients than the activation phase, which could indicate that the binding at the activation site can facilitate the binding at the inhibition site. There is no correlation between the IC50 and EC50 constants and the Hill's coefficients (Figure 1), which indicates that the Hill's coefficients depend on the molecular structure, and not on the binding affinity.

DAPT, semagacestat, and LY-411,575 show quantitative differences in their biphasic dose-response curves (Figure 1), and a very close overlap in 3D structures (Figure 2D). The three compounds have the same peptide backbone, but differ in their flexibility and the molecular volume of the aromatic rings at the C and N termini (Figure 2). LY-411,575 has the lowest EC50 values for activation and IC50 values for inhibition, the highest possible activation levels, and the highest Hill's coefficients (Figure 1). DAPT shows the highest EC50 and IC50 constants with the lower Hill's coefficient and lower maximal possible activation (Figure 1). The values for the biphasic parameters measured with semagacestat are between the values measured with DAPT and LY-411,575.

Avagacestat and LY-411,575 have similar EC50 and IC50, and notable differences in the Hill's coefficients (Figure 1). Avagacestat stands out as a different structure which cannot overlap with the other three molecules in 3D alignments of molecular structures (Figure 2). All four compounds have more than five hydrogen bond donor and acceptor sites, which are surrounded by large hydrophobic surfaces at each end of the molecules (Figure 2). The most notable difference between the four molecules is in the molecular volume (Figure 2), which indicates that the four compounds can bind with different affinity to the different size cavities on the γ-secretase surface.

### 2.2. Multiscale Molecular Dynamics Studies of the Γ-Secretase Structure in Different Steps in the Catalytic Cycle

The flexible parts in the γ-secretase structure can be described using residue-based coarse-grained protocols (Figure 3, [37]). These protocols can routinely depict flexible protein parts in as many as 20 microseconds of molecular events (Figure 3) [37]. We are specifically interested in enzyme–substrate interaction (Figure 3) [38], and in the processing of different Aβ catalytic intermediates (Figure 3) [13,38]. Those complexes are most frequently targeted in different drug development strategies [9,11,13].

Substrate binding and Aβ catalytic intermediates lead to an increase in the width and mobility in the presenilin structure at its cytosolic end (Figure 3A) [38]. This part is exceptionally rich in charged amino acids that form dynamic salt-bridges (Figure 3B, [38], Appendix A). The presenilin structures with Aβ catalytic intermediates show that the different complexes are unstable to a different extent, because the structures with partially-occupied active site tunnels represent a mixture between the *open* and *closed* structures [38,39].

The highest mobility is observed with Aβ 43 and Aβ 46 catalytic intermediates (Figure 3A–C). These highly-mobile structures represent an unstable transient protein-substrate complex in processive catalysis (Figure 3A–B, Appendix A). With Aβ 43 and the shorter Aβ products, the C-terminal end of the substrate forms repulsive interactions with the Asp 257 and Asp 385 in the active site (Figure 3B). This results in a strong negative field which attracts compensating interactions from the positively charged amino acids, most notably Lys 380 and Lys 267 (Figure 3B, Appendix A). The Aβ 46 and longer Aβ products can reach the presenilin interior beyond the active site aspartates (Figure 3B). There, the negatively charged C-terminal makes dynamic salt bridges with different Lys and Arg residues at the cytosolic end of the tunnel (Figure 3B, Appendix A). The presenilin in complex with Aβ 49 does not depend on such large conformational changes to engage in catalysis, and shows lower mobility (Figure 3B,C). The presenilin structure with a full substrate has lower mobility than the no-substrate structure (Figure 3C). The structure with a full substrate depends on the local conformational changes to engage in the catalysis [38]. The no-substrate structure has to support much larger conformational changes in order to bind the substrate and engage in catalysis [38].

When Aβ substrates of different lengths are bound to γ-secretase, the active site tunnel is predominately closed at its cytosolic end (Figure 3B and Appendix A). The active site tunnel can instantly close in molecular dynamics studies when the substrate is removed (Appendix A). The closed structure is necessary to prevent the leaking of the ions across the membrane [38]. The tunnel closing is driven by the dynamic hydrophobic interactions between the amino acids that form the tunnel walls and the flexible protein loops at each end of the tunnel (Appendix A, [38]). The highly dynamic conformational changes drive the different ends of the closed substrate tunnel open to a different extent (Figure 3A). In the longest molecular dynamics simulations, which can depict 20 microseconds of molecular events, in only about 0.4% of the simulation time can two ends of the substrate tunnel be connected. These results indicate that the substrate-analogs or mechanism-based inhibitors can penetrate to the active site aspartates only in a series of consecutive conformational changes.

Root mean square fluctuation (RMSF) graphs [40,41] showed that the highest mobility sites are localized to the flexible loops at the different ends of the active site tunnel, and most notably at the cytosolic end of the presenilin structure (Figure 3). The RMSF graphs showed that high mobility sites overlap with the known drug-binding sites and the hotspots for disease-causing mutations [39,42,43,44,45,46,47] (https://www.alzforum.org/mutations accessed on 2 April 2021). The active site aspartates are part of helix structures, and show low mobility deep in the protein interior; however, they are directly adjacent to the high mobility sites (Figure 3C) [40,41].

### 2.3. Initial Screening for Drug-Binding Sites Using Molecular Docking Studies

Possible binding sites for different drugs can be identified by searching Connolly surface using molecular docking studies (Figure 4, [48]) The docking studies used γ-secretase structures with and without the substrate (Figure 4, [38,39]). Such an approach can describe the competition or cooperation between the substrate and the drugs in binding to the enzyme [9,11]. γ-Secretase is predominately in the substrate-free form in physiological conditions in cells ([11,17] and the references therein).

The presented docking studies identified the same binding sites as the earlier studies which used different experimental techniques (Figure 4, [39,44,45,46,47,49]). The crucial new insight from the presented docking studies is that presenilin can bind multiple drug molecules simultaneously, even in the presence of the substrate (Figure 4). The drugs and the substrate bind at the same sites (Figure 4). Such binding is consistent with the enzyme activity studies, which showed cooperation, competition, or uncompetitive inhibition between the drugs and the substrate at different levels of saturation [9,11].

The drugs bind to the most mobile parts in the presenilin structure (Figure 3 and Figure 4), which regulate the opening of the 29 Å long active site tunnel [38,39], and thus must have effects on substrate binding and processive catalysis [13,27]. The binding sites for the drugs overlap with the hotspots for many of the disease-causing FAD mutations (Figure 4, https://www.alzforum.org/mutations accessed on 2 April 2021 [38,39]). The overlapping structural elements are in line with the activity studies which showed that the drugs and FAD mutations can produce similar changes in γ-secretase activity [7,11,13].

The possibility that several drug molecules can bind to γ-secretase simultaneously was suggested in some of the earlier studies [45]. Multiple studies also indicated that the same drugs can sometimes bind to different sites [39,44,45,46,47,49]. However, to our knowledge, the functional consequences of multiple enzyme–drug interactions have been explored only in our mechanistic studies of biphasic dose-response [11,12].

### 2.4. All-Atom Molecular Dynamics Studies Of Binding Interactions Between Biphasic Drugs and γ-Secretase in The Presence and Absence of A Substrate

We have combined molecular docking studies (Figure 4) and molecular dynamics studies (Figure 3). γ-Secretase structures with one, two, or three drug molecules bound to the enzyme are shown in Appendix A, were used to describe the ways in which drugs can affect the dynamic structures that regulate enzyme–substrate interactions [38,39,49]. We found that all four drugs can decrease the mobility of the flexible protein loops that regulate the opening of the active site tunnel (Figure 3 vs. Figure 5 and Figure 6). The loops control the release of tripeptide catalytic byproducts, and the processive catalysis (Figure 3 and [7,13,27]). With all four drugs, the binding constants in the cell-based assays (Figure 1) correlate with the observed RMSF values (Figure 5 and Figure 6, panel D). The biggest decrease in the RMSF mobility values shows the compounds that have the highest binding affinities, the biggest volume, and the lowest flexibility (Figure 1 and Figure 2). Thus, the cell-based and in-silico studies show consistent results (Figure 1 vs. Figure 5 and Figure 6).

We found that the drugs bound at the different ends of the active site tunnel could mimic conformational changes in the enzyme structure that drive the release of tripeptide byproducts and processive catalysis (Figure 3B and [7,13,27]). Driven by the flexible loops (Figure 3A and Appendix A), the drugs slide into the cavities on the protein surface. The drugs can even penetrate into the active site tunnel towards Asp 257 and Asp 384 when the enzyme is in the substrate-free form (Figure 6, Appendix A and Appendix A). These motions can be quantitatively described for each drug using RMSD values (Figure 5 and Figure 6, panels C). A comparative analysis of the drugs bound at both ends of the active site tunnel, or only at one end of the tunnel, showed that the drugs can penetrate faster and deeper into the tunnel when bound at both ends (Appendix A and Appendix A). Such cooperativity is consistent with the high Hill’s coefficients that were observed in the binding assays (Figure 1). The cooperative binding is also in agreement with other binding studies, which indicated that the binding of one drug molecule can facilitate the binding of another drug molecule [11,45].

The binding site at the cytosolic end of the presenilin structure is the most dynamic (Figure 3A, Figure 5 and Figure 6). In the presence of the substrate, the substrate forces the drugs away from TM6a and TM7 towards TM8 and Aph1 (Figure 5 and Appendix A). In the absence of the substrate, the flexible protein loops first drive the drug into the position of the substrate between TM6, TM6a, and TM7 (Figure 6 and Appendix A). From there, the drugs can bind between TM6 and TM7 and slide toward the active site Asp 257 and Asp 385, or bind between TM7 and TM8 and slide towards the PAL motif (a.a. 433–436, [42]). The binding of the first drug molecule can facilitate the binding of the second drug molecule by opening the protein loops at the adjacent sites. The drugs can bind and slide between the loops that form an 18 Å long surface from TM6 to TM8. The sliding drugs form mostly-hydrophobic interactions with the dynamic cavities on the presenilin surface, and up to three highly-dynamic hydrogen bonds with the flexible protein loops or the surrounding water molecules. The most extensive sliding is observed with DAPT and semagacestat; the lowest sliding is observed with LY-411,575 and avagacestat (Figure 5 and Figure 6, Appendix A, and Appendix A). The binding of multiple drug molecules can block the drifting within the drug-binding cavities, and thus multiple drug molecules compete more effectively with the conformational changes in the flexible protein loops (Figure 6C).

The second binding site, at the membrane end of the active site tunnel, can bind only one molecule with each of the four drugs [39] (Figure 5 and Figure 6). In the absence of the substrate, the drugs form contacts with TM2, TM3, and TM5 (Figure 6A,B). The mobile linkers between TM2, TM3, and TM5 (Figure 3, residues 110 to 130, and 214 to 217) control the width of the cavity and the penetration of the drugs into the active site tunnel (Figure 6, Appendix A and Appendix A). Buried in a predominantly hydrophobic cavity, the drugs initially form several hydrophobic interactions, most notably with Ile 138 and Val 142 on TM2. Phe 237 and Tyr 115 can form π–π stacking interactions with the terminal aromatic parts of each drug (Appendix A). The inhibitors can also form dynamic hydrogen bonds with the side chains on Thr 44 on the substrate, and with Tyr 115 on presenilin ( Appendix A). The site is dominated by hydrophobic interactions. In less than 10% of molecular dynamic conformations, the drugs form one hydrogen bond with the protein ( Appendix A). The substrate can displace the interactions between the drugs and TM3 and TM5 (Figure 5A,B, Appendix A). The drug is burred in a tight hole between TM2, Gly 38 and Val 40 on the substrate (Figure 5, Appendix A). These observations are in agreement with the studies that indicated that the drugs can form nonspecific interactions with the substrate and affect Aβ production [52].

The third binding site is in the hydrophobic gap between the Aph1 subunit and the TM8 in presenilin 1. Any analysis at this site has to be taken with some reservations, since that part of the structure lacks 73 amino acids on presenilin 1 [7,13,27]. Nevertheless, the site is interesting because different Aph1 subunits can affect the production of toxic Aβ products [53], with no effect on the overall turnover rates of γ-secretase [53]. The underlining mechanism is still unknown [53]. This site showed the lowest RMSD values for all four drugs, and the lowest influence by the substrate (Figure 5 and Figure 6). We found that the drugs facilitate the breaking of the contacts between TM8 on presenilin, and TM2 and TM3 on Aph1 (Figure 5 and Figure 6 and Appendix A). The drugs are buried in a hydrophobic gap between Leu 420, Leu 423, Ile 427 on TM8 on presenilin, and Leu 93, Leu 96 on TM2 of Aph1 (Appendix A). The dynamic conformational changes can drive the gap between open and closed conformation. The closed conformation is stabilized by a salt bridge between Glu367 at the endo-proteolytic site and Lys 429–430 at the cytosolic end of TM8.

### 2.5. All-Atom Molecular Dynamic Studies of γ-Secretase Structure With Aβ Catalytic Intermediates in Processive Catalysis

The structures of γ-secretase in complex with different Aβ catalytic intermediates were used for the analysis of the modulation of Aβ production by biphasic drugs [7,9,13,27] (Figure 7 and Appendix A). Earlier studies of biphasic dose-response with DAPT showed that the drugs can facilitate the production of shorter Aβ products at the lower drug concentrations, and the production of longer Aβ peptides at higher drug concentrations [13]. Those observations indicated that, with different Aβ catalytic intermediates, the drugs bind at two different sites: a higher binding affinity site with the shorter Aβ products, and a lower binding affinity site with the longer Aβ products.

We prepared γ-secretase in complex with Aβ 40, Aβ 43, Aβ 46, and Aβ 49 for the analysis of the binding interactions with all four biphasic drugs (Figure 7). The structures with Aβ 40, Aβ 43, Aβ 46, and Aβ 49 represent—to different extents—a mixture of *open* and *closed* structures of γ-secretase (Figure 3A and [38,39]). When Aβ substrates of different lengths are bound in the active site tunnel, the tunnel is predominately closed at its cytosolic end (Figure 7, Appendix A). Thus, we started the simulations by docking the drugs at the cytosolic end of presenilin with the active site tunnel closed.

The drugs penetrate into the active site tunnel in the presence of Aβ catalytic intermediates (Figure 7), driven by the same mechanism that drives the penetration of multiple drug molecules at different ends of the tunnel (Figure 6, Appendix A and Appendix A). Aβ molecules can facilitate the opening of the tunnel and the penetration of the drugs into the tunnel (Figure 7) faster than the drug molecules that bind at the membrane-embedded end of the tunnel (Figure 6, Appendix A and Appendix A). The drugs that penetrate into the active site tunnel bind between flexible loops that can selectively disrupt the processive cleavages of the nascent Aβ peptides (Appendix A and [7,9,13]). Different binding interactions, and different RMSF values, are in agreement with the earlier observations which showed that the modulation of Aβ production by different drugs depends on the structure and concentration of each drug [7,9,13,27].

With short Aβ 40 and Aβ 43, the drugs can penetrate deep into the active site tunnel and show the lowest RMSF values when bound inside the tunnel (Figure 7). The drugs bind between amino acids 146 to 173 and 226 to 387 in the predominantly-hydrophobic surface, which can form only up to one dynamic hydrogen bond (Figure 7). Parts of the binding surfaces are Pro284, Ala285, and Leu286, and the aromatic amino acids Tyr154, Trp165, and Phe283 (Figure 7). Aromatic residues can contribute to π–π stacking. The penetration depends on the molecular structures. DAPT and LY-411,575 have an elongated flexible structure which can line up with the tunnel walls and penetrate deeper into the channel. A lower depth of penetration is observed with semagacestat. Semagacestat has structural similarities with DAPT and LY-411,575, but its small structure has much smaller effects on flexible loops in the presenilin structure. Avagacestat has a wide structure which cannot line with the tunnel walls and penetrate deep into the tunnel.

The longer Aβ 46 and Aβ 49 catalytic intermediates do not allow the deep penetration of the drugs into the active tunnel (Figure 7). The drugs bind in the hydrophobic cavities, where they can form up to three dynamic hydrogen bonds with the charged amino acids on dynamic flexible loops (Figure 7, Appendix A). The drugs bind between amino acids 73 to 83, 378 to 381, and 417 to 435 in the active site loop. Part of the binding site is the PAL motif (Pro433, Ala434, Leu435, Pro436) [42], and aromatic amino acids Tyr77 and Phe428, which can result in π–π stacking interactions with the drug (Figure 7).

## 3. Discussion and Conclusions

We combined structure and activity studies of the biphasic dose-response drugs that target γ-secretase (Figure 1 and Figure 2 and [7,9,11,12,13,16,17]). The dynamic descriptions of molecular interactions are presented at 37 °C, in a physiologically-relevant lipid bilayer, using the available cryo-Electron Microscope (EM) structures [38]. The presented structural results support our earlier enzyme activity studies [11,12,17]. Both the structure and the activity studies consistently showed that multiple drug molecules can bind to γ-secretase in parallel. The biphasic dose-response activities are a result of the selective action of drugs in different steps in the catalytic cycle (Figure 3, Figure 4, Figure 5, Figure 6 and Figure 7, Appendix A and Appendix A).

### 3.1. *Biphasic Drugs: The Activation Mechanism*

The activity studies showed that the biphasic drugs can activate γ-secretase by facilitating enzyme-substrate interactions in conditions with limiting substrates [11,12]. Presented structural studies showed that the drugs can facilitate enzyme–substrate interactions by two mechanisms. The drugs can facilitate the opening of the active site tunnel at each end (Figure 6, Figure 7, and Appendix A, and Appendix A), and the drugs can stabilize the flexible loops that control the formation of the enzyme-substrate complex (Figure 5 and Appendix A). The active site tunnel is tightly closed in the absence of the substrate in order to prevent water and ions from leaking across the membrane [38,39] (Figure 3, Appendix A). The opening of the tunnel is the rate-limiting step in the process of the initial enzyme–substrate recognition [38,39] (Figure 3). The drugs can facilitate the opening of the tunnel, and thus activate γ-secretase when the tunnel opening and enzyme–substrate recognition are the rate-limiting steps [11,12,17].

The active site tunnel is closed in the absence of a substrate due to hydrophobic interactions inside the tunnel, and due to flexible protein loops at the ends of the transmembrane helixes (Figure 3, Appendix A) [38,39]. The drug molecules can bind to the loops and force the helixes apart by mimicking the substrate in the process of enzyme–substrate recognition, or by mimicking the release of tripeptide catalytic products (Figure 3, Figure 6 and Figure 7, Appendix A, and Appendix A). The highest activation was observed with LY-411,575, which has the strongest binding affinity (Figure 1) [36]. LY-411,575 has an elongated, bulky structure (Figure 2) which can penetrate deep into the tunnel (Figure 6 and Appendix A). Its bulky head can form π–π stacking interactions (Appendix A) and stabilize the presenilin structure in open conformations [38,39]. Looking at known drugs that target soluble aspartic proteases, we found that the active site tunnel and the flexible protein loops on the cytosolic side (Figure 3) resemble the functions of the active site loops that are usually targeted in drug development with this type of enzyme [54,55].

### 3.2. Biphasic Drugs: The Inhibition Mechanism

The presented structures can also explain the surprising uncompetitive inhibition that is observed with biphasic drugs [11,12,17]. The uncompetitive inhibitors are not acceptable in any of the drug-development strategies with γ-secretase as the target. At lower doses, the uncompetitive inhibitors can produce the same changes in γ-secretase activity as FAD mutations in presenilin 1 [22,26], and could possibly facilitate pathogenesis [11,17]. At higher doses, the uncompetitive inhibitors can stop the vital functions of γ-secretase in cell physiology [2,5,56]. The uncompetitive inhibition is surprising because the drugs were designed to mimic the substrate and catalytic intermediates, and to bind in the active site tunnel in the place of the substrate. The presented results showed that the drugs bind to flexible protein parts that drive processive cleavages and the release of tripeptide intermediates (Figure 3, Figure 5, Figure 6 and Figure 7, and Appendix A). The uncompetitive inhibition [11,12,17] is a result of interference with the processive catalytic steps, which are frequently associated with pathogenesis [13,17,27].

More importantly, it is often forgotten that the activity studies showed that different Aβ products show biphasic dose-response at different drug concentrations [13]. The shorter Aβ products show activation at the lower drug concentrations, while the longer Aβ products show activation at the higher drug concentrations [13,27]. Thus, there are high-affinity binding sites for the drugs with the shorter Aβ peptides, and the lower affinity binding sites for the drugs with the longer Aβ peptides. In line with those observations, we found that the drugs can penetrate deeper into the presenilin structure with the shorter Aβ peptides (Figure 7). The drugs can mimic tripeptide catalytic products in the processive cleavages (Figure 3 and Figure 7, Appendix A). The depth of penetration, and consequently the binding affinity, depends on the ability of each drug to jam the flexible loops that regulate processive catalysis and the tripeptide release (Figure 3 and Figure 7, and Appendix A). It is very likely that all of the studies that have reported the modulation of Aβ production, or selectivity between APP vs. Notch substrate, had the biphasic response mechanism [9,11,13]. The biphasic mechanism was not always detected because the measurements used a limited concentration range for the drugs and the substrate [9,10,11].

### 3.3. Biphasic Drugs and FAD Utations

Mechanistic similarities between biphasic drugs and FAD mutations are very important because they can be exploited in future studies of disease pathogenesis [7,11,13,17,22,25,26,32,57,58,59,60,61,62,63,64]. Biphasic drugs can be used like FAD mutations in studies of pathogenic changes in the amyloid metabolism in cell cultures and model animals [25]. Presented results showed that the drugs and the FAD mutations target the same dynamic parts in presenilin structures (Figure 4, Figure 5, Figure 6 and Figure 7, Appendix A, and https://www.alzforum.org/mutations accessed on 2 April 2021, [4,38,39]). FAD mutations are known to affect the biphasic dose-response [17] and the binding constants for the drugs [22]. Biphasic drugs and FAD mutations favor an increase in the production of the longer, more hydrophobic Aβ catalytic intermediates [7,9,11,13,16,22,26,27]. Presented results indicate that the increase can be attributed to the selective interference with the dynamic conformational changes at the cytosolic end of TM6, TM6a, and TM7 (Figure 7, Appendix A).

### 3.4. Concluding Remarks on Future Drug Development Strategies with γ-Secretase as the Target Enzyme

The insights from mechanistic studies of biphasic dose-response can help to achieve reproducible results and sustained progress in future drug-development efforts [11,13,18,19,23,27,32].

The majority of preclinical drug development studies did not pay attention to the extent of γ-secretase saturation with its substrate, and most often used a saturating substrate. The assays with saturating substrate are the fastest and the cheapest, but also the least relevant to cell physiology [7,9,10,16]. In cells, in physiological conditions, γ-secretase is far from saturation with its substrate [9,10,17]. The sub-saturated enzymes are a fundamental physiological mechanism for all enzymes [21], which can assure the fastest and best-controlled response to metabolic fluctuations [18,19,20]. Artificial preclinical studies with γ-secretase at saturating substrate have limited clinical relevance, and often lead to misleading conclusions [7,9,10,16]. Measurements with a poorly-defined saturation of γ-secretase with its substrate can explain why so many studies reported inconsistent results on the modulation of Aβ production, and/or on selectivity between different substrates [5]. Different substrates and different Aβ products have different Kӎ values [13,18,22,23]. Kӎ values are a crucial parameter for a meaningful description of preferences for different substrates and Aβ products [18,19,22,23]. The activity assays with saturating substrate can also favor uncompetitive inhibitors, which can produce the same changes in the γ-secretase activity as FAD mutations [11,17,22,26]. The activity assays with saturating substrate failed to observe the biphasic activation–inhibition dose-response curves [9,10,17].

The biphasic dose-response curves can give a false impression that the drugs can be used at precise doses to act only as activators or only as inhibitors of γ-secretase (Figure 1) [10,15,16]. Such targeted dose therapy is not possible. Both the activation and the inhibition by biphasic drugs can decrease the catalytic capacity of γ-secretase, just like the activating and the inhibiting FAD mutations [11,17]. The decrease in catalytic capacity can be a result of the increase in enzyme saturation with its substrate, or a decrease in the turnover rates, or both [11,17,18]. A wide range of diverse studies showed a good correlation between a decrease in γ-secretase capacity to process its substrates and the pathogenic events [17,32,57,58,59,60,61,62,63,64,65]. Based on those observations, we are proposing that competitive inhibitors of γ-secretase have the best chance to become drugs for Alzheimer’s disease [17,18,19]. Similar to the competitive inhibitors, the protective A673T mutation in the APP substrate can decrease the extent of the enzyme saturation with its amyloid substrate, with no effects on the turnover rates [61]. The aim is not to inhibit γ-secretase. The aim is to modulate the extent of the enzyme saturation with its substrates in correlation with changes in the metabolic load for APP [3] and Notch substrates [17,18]. The development of competitive inhibitors requires changes in the drug-screening strategies [18,19].

In conclusion, numerous and expensive failures of different drug-design strategies are not evidence that γ-secretase is not a good therapeutic target, but are rather evidence that ‘fast-and-cheap’ protocols have overlooked the key features of this uniquely complex protease. The drug-screening strategies must follow fundamental protocols for the analysis of enzyme activity [18,19,24] in order to achieve reproducible results and sustained progress [11,13,18,19,23,27,32]. The failed clinical trials with semagacestat [7,16,30,31,66] were much more expensive than the failed clinical trials with avagacestat [10,28]. In a large part, that can be attributed to better documented—and more comprehensive—preclinical studies with avagacestat [9,10,15,28,29].

## 4. Materials and Methods

### 4.1. Chemicals

Drugs were purchased from Calbiochem: DAPT (N-[N-(3,5-difluorophenacetyl)-l-alanyl]-S-phenylglycine t-butyl ester). Semagacestat, LY-450,139 2S)-2-Hydroxy-3-methyl-N-[(1S)-1-methyl-2-oxo-2-[[(1S)-2,3,4,5-tetrahydro-3-methyl-2-oxo-1H-3-benzazepin-1-yl]amino]ethyl]-butanamide; LY-411,575, N2-[(2S)-2-(3,5-Difluorophenyl)-2-hydroxyethanoyl]-N1-[(7S)-5-methyl-6-oxo-6,7-dihydro-5H-dibenzo[b,d]azepin-7-yl]-L-alaninamide; Avagacestat, (2R)-2-[N-[(4-Chlorophenyl)sulfonyl]-N-[[2-fluoro-4-(1,2,4-oxadiazol-3-yl)phenyl]methyl]amino]-5,5,5-trifluoropentanamide, (R)-2-(4-Chloro-N-(2-fluoro-4-(1,2,4-oxadiazol-3-yl)benzyl)phenylsulfonamido)-5,5,5-trifluoropentanamide, BMS-708163.

### 4.2. Secretion of Aβ 1-40 in Cultures of SH-SY5 Cells in the Presence of Increasing Concentrations of Drugs

SH-SY5 cells were purchased from ATCC as passage 11, and were maintained in Dulbecco’s modified Eagle’s medium (DMEM) supplemented with 10% fetal bovine serum. The measurements of the biphasic response in the presence of the drugs, and the corresponding data analysis were described in detail in our earlier studies [11,17]. Briefly, different concentrations of drugs were prepared in DMSO, and were added to the cells so that the final DMSO concentration in the culture was 0.1% (*v/v*). The DMSO vehicle represents 0 nM drugs. The cells were incubated in 6-well plates, with the drugs at the given concentrations, for between 12–18 h. We paid maximal attention to measure all four drugs under identical conditions. The same batch of cells was used in parallel for the measurements with all four drugs. Identical conditions were used for the incubation of the drugs with the cells, sample harvesting, and measurements of Aβ 1-40.

### 4.3. Sandwich ELISA for Quantitative Detection of Aβ 1-40 

The assays closely followed the manufacturer’s instructions. Briefly, sandwich ELISA kits for the quantitative detection of human Aβ 1-40 peptides in a flexible 96-well format were purchased from Millipore (cat. #. TK40HS, The Genetics company Switzerland). The assay had a linear response in the range from 6 to 125 pM of Aβ 1–40. The Aβ 1–40 samples from the cell cultures were used immediately after their collection, following the manufacturer suggestion and our earlier reported experimental experiences [11]. The wells were filled with 50 µL of the antibody conjugate solution and 50 µL of the sample. The Aβ 1–40 standards supplied by the manufacturer were prepared in parallel with the other samples. The prepared wells were wrapped in aluminum foil and incubated overnight at 4 °C with gentle mixing. The next day, each well was washed five times with 300 µL wash solution. After each 20 min wash, the wash solution was poured out and the wells were dried by tapping the plates on an absorbing paper. After washing, the wells were filled with 100 µL of the enzyme conjugate solution, covered, and incubated for 30 min at room temperature, with shaking. The washing procedure was repeated once again, in order to remove the excess of the enzyme-conjugate. Next, 100 µL of the substrate solution was added to each well in the dark, and kept for 30 min, covered, at room temperature. The reaction was quenched by adding 50 µL stop solution to each well, and within 15 min the signal intensity was read by measuring absorption at 450 nm.

### 4.4. Inhibitor Docking Studies

The binding sites for semagacestat, avagacestat, LY-411,575, and DAPT were calculated following the standard protocols [48,67]. Briefly, the ligands were hydrogenated and charged using the Gasteiger protocol and pH = 7.0. The proteins were protonated at pH = 7.0 using AMBER98S force filed with Asp 257 and Asp 385 unprotonated.

### 4.5. Residue Basedcoarse-Grained Molecular Dynamics Studies

Coarse-grained molecular dynamics calculations with the γ-secretase structure within the lipid bilayer were prepared using CHARMM-GUI 32 Martini Bilayer Maker 33 [37,68]. The OPM protocol was used to position and orient the proteins in a lipid membrane bilayer [69]. The systems were relaxed using equilibration steps with the temperature set to 310 K using V-rescale coupling, and the pressure was set to 1.0 bar using semi-isotropic Berendsen coupling [51]. The systems with a mixed lipid bilayer had 1355 residues, 1604 lipid molecules, 71,112 water molecules, 924 Na+ ions, and 791 Cl- ions in a 210 Å × 210 Å × 246 Å box. In order to ensure proper system relaxation, one minimization step and five equilibration steps were used. The lipid composition of the membrane bilayer in the CG systems [70] (lipid type, number of lipid molecules, percentage of lipid molecules) was: phosphatidylcholine (POPC), 340, 21%; phosphatidylethanolamine (POPE), 176, 11%; phosphatidic acid (POPA), 16, 1%; phosphatidylserine (POPS), 64, 4%; sphingomyelin (PSM), 96, 6%; phosphatidylinositol (POPI), 32, 2%; cholesterol (CHOL), 880, 55%. The lipid composition is crucial for the achievement of a presenilin structure with proper orientation between the active site aspartates ([71] and (Svedružić et al. in preparation). The γ-Secretase structures in molecular dynamics simulations were depicted in time-steps of 20 fs in order to describe periods of 10 to 20 µs of molecular events in 1000 to 2000 frames. Calculations used between 480 to 1440 processors on Atos, Bullx DLC 720, Xeon E5-2690v3 12C 2.6GHz, 288 nodes,288 nodes with 48 processors and 64 GB per node, Infiniband FDR systems from 48 to 120 h.

### 4.6. All-Atom Molecular Dynamics Studies

γ-Secretase structures in complex with different drugs from the molecular docking studies [48,72] were used as the starting structures for the molecular dynamic studies. γ-Secretase structures with the active site tunnel closed were prepared in the molecular docking studies using the structures without the substrate as the starting structures.

The all-atom simulations of γ-secretase in the lipid bilayer were prepared using CHARMM-GUI Membrane Builder 35–37 [73]. OPM protocols were used to position and orient the proteins in a lipid membrane bilayer [69]. The systems were relaxed using a sequence of equilibration steps, with the temperature set to 303.15 K using Nose–Hoover coupling, and the pressure was set to 1.0 bar using semi-isotropic Parinello–Rahman coupling. The systems used in our simulations contain 85,080 TIP3 water molecules, 272 potassium POT ions, 221 chlorine CLA ions, and 708 lipid molecules. The overall number of atoms in these systems is 347,516, and they are contained in a 144 Å × 144 Å × 176 Å box. The temperature of the simulation was set to 303.15 K using Nosé–Hoover coupling. The pressure was set to 1 bar using semi-isotropic Berendsen coupling. The constraint algorithm was LINCS, and the cut-off scheme was Verlet. In order to ensure proper system relaxation, two minimization steps and four equilibration steps were used. The simulations analyzed molecular processes from 300 to 600 nanoseconds on a molecular time scale, with a 2 fs time step. The lipid composition in the membrane bilayer in all-atom systems (lipid type, number of lipid molecules, percentage of lipid molecules): phosphatidylcholine (POPC), 152, 21%; phosphatidylethanolamine (POPE), 78, 11%; phosphatidic acid (POPA), 8, 1%; phosphatidylserine (POPS), 28, 4%; sphingomyelin (PSM), 42, 6%; phosphatidylinositol (POPI), 14, 2%; cholesterol (CHOL), 386, 55%.

The validities of the presenilin structures in each of the molecular dynamic simulations were analyzed by comparing the active site structures with the mechanistic studies of the active site of aspartic protease [71], and by comparing the calculated and experimental pKa for the active site aspartates [50,74] (Svedružić et. al. manuscript in preparation).

The ligand parameterization was prepared using ACPYPE tools [75] or the CHARMM-GUI Ligand Reader & Modeler, in order to calculate the CHARMM-compatible topology and parameter files [76]. The all-atom input files with γ-secretase in the lipid bilayer were prepared using the CHARMM-GUI Membrane Builder [77]. The EM refined structures of γ-secretase (PDB: 6IYC) [38] had a total of 1355 residues, five chains, and a resolution of 2.60 Å. The proteins were positioned and oriented in the lipid membrane bilayers using the OMP protocol [69]. The molecular structures for semagacestat, avagacestat, LY-411,575, and DAPT were taken from the ChemSpider and PubChem databases. All of the simulations used GROMACS version 2019.4 [51]. The molecular dynamics for γ-secretase in complex with different drugs were depicted in time-steps of 2 fs, in order to represent a total of 300 to 600 nanoseconds of the molecular events in 100 to 200 recorded frames. Calculations used between 1440 to 2400 processors on Atos, Bullx DLC 720, Xeon E5-2690v3 12C 2.6GHz, 288 nodes with 48 processors and 64 GB per node, Infiniband FDR systems for 48 to 120 h.

### 4.7. Data Analysis and Presentation

Molecular imaging and analyses of the molecular properties used VMD 1.9.3 and UCSF Chimera 1.14 [72,78]. The molecular trajectories were analyzed using Bio3D package with R 3.6.2 and Rstudio 1.2.5019's [40,41]. The RMSF (root-mean-square-fluctuations) values were calculated as a function of molecular time [51] or the position of amino acid residues [40,41]. All of the biphasic profiles were analyzed using nonlinear regression, and the equation for the biphasic dose-response curve that was described in detail in our earlier studies [11,17]:(1)$$S\left(x\right)\;=\;\mathrm{PA}\;+\;\frac{\left(MA-IA\right)}{\left(1+10^{\left(EC50-X\right)\cdot p}\right)}+\frac{\left(MA-MI\right)}{\left(1+10^{\left(X-IC50\right)\cdot q}\right)}$$
where, *S(x)* represents the measured activity at inhibitor concentration *x*. *PA* is the physiological Aβ 1–40 production activity at inhibitor concentration zero. *MA* is the calculated maximal activity caused by activation if there is no competing inhibition. The *MA* values roughly correlate with the capacity of drugs to induce an increase in the enzyme–substrate affinity. *MI* is the maximal inhibition. *EC50* and *IC50* represent the activation and inhibition constants, respectively, while *p* and *q* represent the corresponding Hill’s coefficients.

## Figures and Tables

**Figure 1 pharmaceutics-13-00514-f001:**
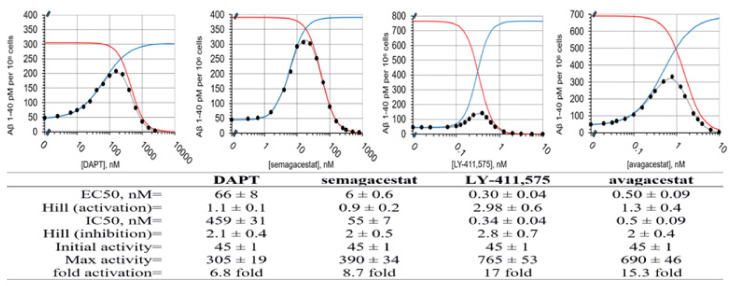
Dose-response curves for Aβ 1-40 production in SH-SY5 cells in the presence of DAPT, semagacestat, LY-411,575, and avagacestat. The Aβ 1-40 production in the SH-SY5 cells shows biphasic activation–inhibition dose-response curves with all four drugs (●). The different parameters that describe the biphasic dose-response curves were calculated and listed in the table (methods, Equation (1)) [11]. The gray lines represent the best-fit curve to the experimental values (●). The blue and red lines represent calculated activation and inhibition events if the two events can be separated (methods, Equation (1)). The activation constants (EC50) and the inhibition constant (IC50) represent the affinity for each binding event. The Hill's coefficients represent the stoichiometry of the interaction, and/or possible cooperative processes in the binding events. The ‘Max activity’ parameter represents the maximal possible activation if there is no competing inhibition. The ‘Max activity’ parameter correlates with the ability of drugs to facilitate enzyme–substrate interactions [11]. The initial activity is the same for all four drugs, because all of the measurements used the same batch of cells.

**Figure 2 pharmaceutics-13-00514-f002:**
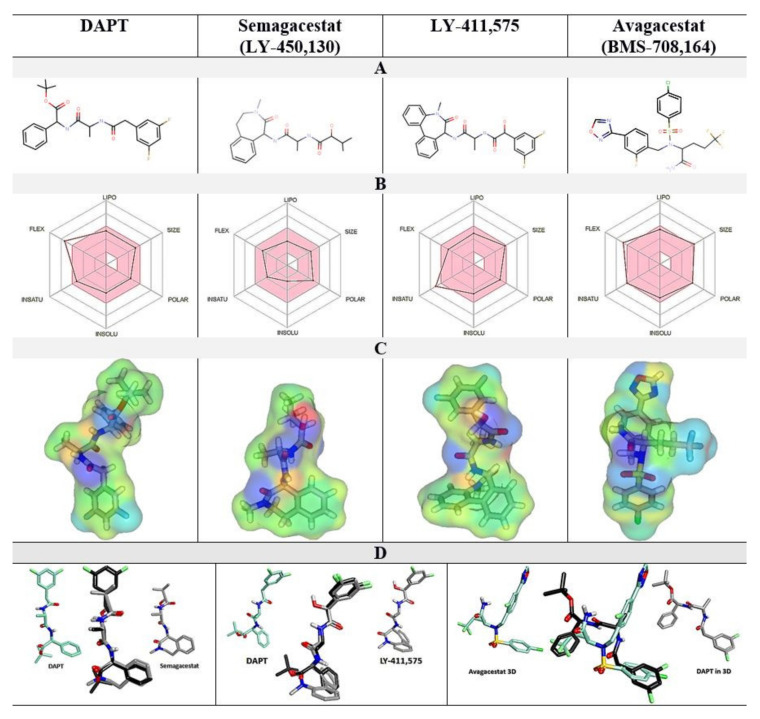
(**A**–**D**). Comparative analysis of the physicochemical properties of DAPT, semagacestat, LY-411,575, and avagacestat structures. (**A**) 2D structures. (**B**) A radar diagram shows the overlap with the Lipinski rules: FLEX flexibility, INSAT relative share of sp^3^ carbon atoms, INSOLU LogP values, POLAR polar surface area, SIZE molecular mass, and LIPO hydrophobic surface area. The pink area represents the optimal values; the superimposed lines represent the values specific to each compound. (**C**) The electron densities are mapped on the molecular surfaces and colored to highlight the surface properties: green, hydrophobic; blue, H-bond donor; red, H-bond acceptor; yellow, polar. (**D**) The overlap between 3D molecular structures and DAPT as the reference molecule.

**Figure 3 pharmaceutics-13-00514-f003:**
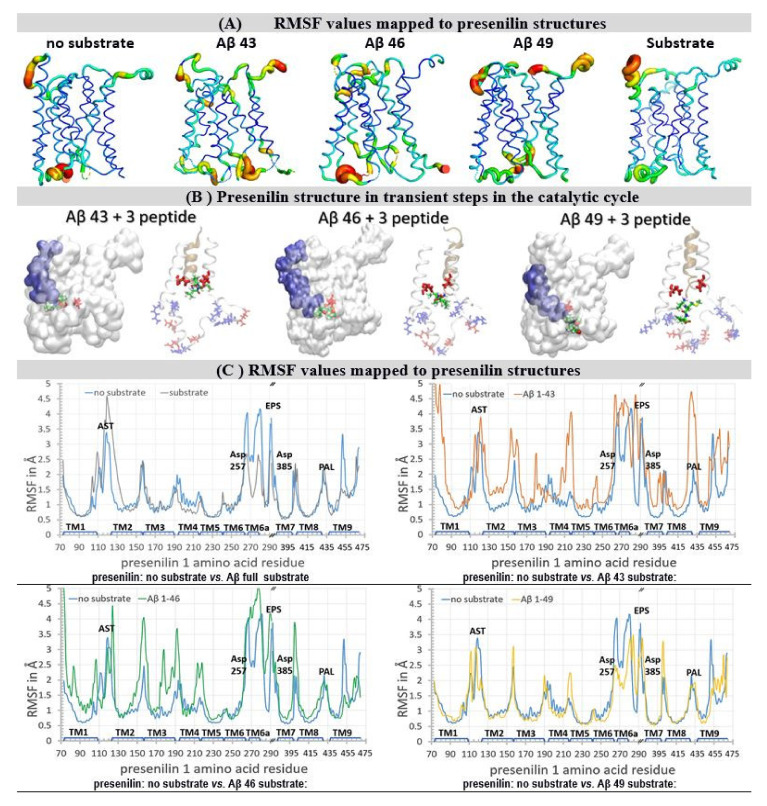
(**A**–**C**) Residue-based coarse-grained molecular dynamics studies of the γ-secretase complex. We used cryo-EM coordinates (PDB:6IYC) [38] to describe how the substrate and Aβ catalytic intermediates can affect the dynamic conformational changes in the γ-secretase complex. The described conformation changes represent between 10 to 20 µsec of molecular events [37]. The results are depicted visually (**A**,**B**), and quantitatively (**C**), with the focus on presenilin 1 structures. (**A**) Principal components analysis can depict the relative differences in protein mobility [40,41]: thin blue lines represent the lowest mobility, green and yellow lines represent intermediate mobility, and thick red lines represent the highest mobility. The most mobile parts are the two ends of the active site tunnel, and at the C-terminal part that penetrates in the Aph1 subunit. The substrate and Aβ catalytic intermediates can predominately affect the mobility at the cytosolic end of the active site tunnel. These motions are consistent with the processive cleavages, and release of the three peptide byproducts. (**B**) The presenilin structures with Aβ catalytic intermediates show that different complexes are unstable to a different extent, because the structures with a partially-occupied active site tunnel represent a mixture between the *open* and *closed* structures [38,39]. The structures are shown as a white transparent surface to make the structures underneath the surface visible. Different Aβ catalytic intermediates in the active site tunnel are depicted as blue surfaces, active site Asp 257 and Asp 385 are shown as red licorice. The white ribbon models depict amino acids 240 to 394 in presenilin structures, while the gold ribbons depict different Aβ catalytic intermediates. The positively-charged amino acids are shown as blue licorice; negatively charged amino acids are shown as red licorice, including the active site Asp 257 and Asp 385. The tripeptide byproducts of processive catalysis are shown as green models [13,27,38]. (**C**) Root mean square fluctuation (RMSF) values give a quantitative description of the structural changes at the level of each amino acid [40,41]. The biggest variability in the RMSF values is observed in the structural parts that drive the enzyme–substrate interaction and the processive catalysis [13,27,38]. The key structural elements are mapped on the graph [38]: TM1–TM9, Trans Membrane helix 1 to 9; AST, membrane-embedded opening of the Active Site Tunnel; EPS, Endo-Proteolytic Site; PAL, motif (Pro 435, Ala 434, Leu 433) [42]; Asp 257 and Asp 385 in the active site.

**Figure 4 pharmaceutics-13-00514-f004:**
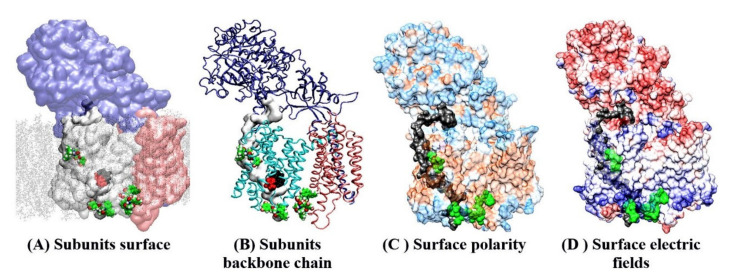
(**A**–**D**) Docking sites for semagacestat on the γ-secretase structure in a complex with its substrate (PDB:6IYC) [38]. Molecular docking studies showed that as many as four semagacestat molecules (green) can bind to γ-secretase simultaneously in the presence of the substrate. One drug molecule binds at each end of the active site tunnel; two drug molecules are buried in the gap between presenilin and the Aph1 subunit. Transparent protein surfaces are used to show the structures buried in the protein interior. Four different presentations show the position of the drug-binding sites relative to the main structural elements in the γ-secretase complex. (**A**,**B**) Different subunits are shown: nicastrin, cyan; presenilin, white or light blue; Aph1, pink; substrate, gray; the membrane is shown as dots. The drugs bind to presenilin sites inside and outside of the membrane. Buried underneath the protein surface is the substrate (depicted as a gray surface), the active site Asp 257 and Asp 385 (red), and the adjacent PAL motif (black, Pro 435, Ala 434, Leu 433) [42]. (**C**) The protein surface is colored based on its polarity: blue, polar; brown, hydrophobic; white, amphiphilic. The substrate is shown as a black surface, and it marks the position of the active site tunnel. (**D**) Adaptive Poisson-Boltzmann Solver APBS analysis of electric fields on the protein surface: blue, positive; red, negative; white, neutral [50]. The electric fields show that γ-secretase is a polarized molecule. The negative field dominates on the nicastrin side of the membrane, while the positive field dominates the cytosolic site. The positive N-terminal of the substrate matches the negative field on nicastrin. The negative C-terminal on the nascent Aβ catalytic intermediates matches the positive field at the cytosolic side of the protein. Dynamic electric fields can be a crucial part of enzyme–substrate recognition and the processive cleavages of the Aβ catalytic intermediates [13,27,38].

**Figure 5 pharmaceutics-13-00514-f005:**
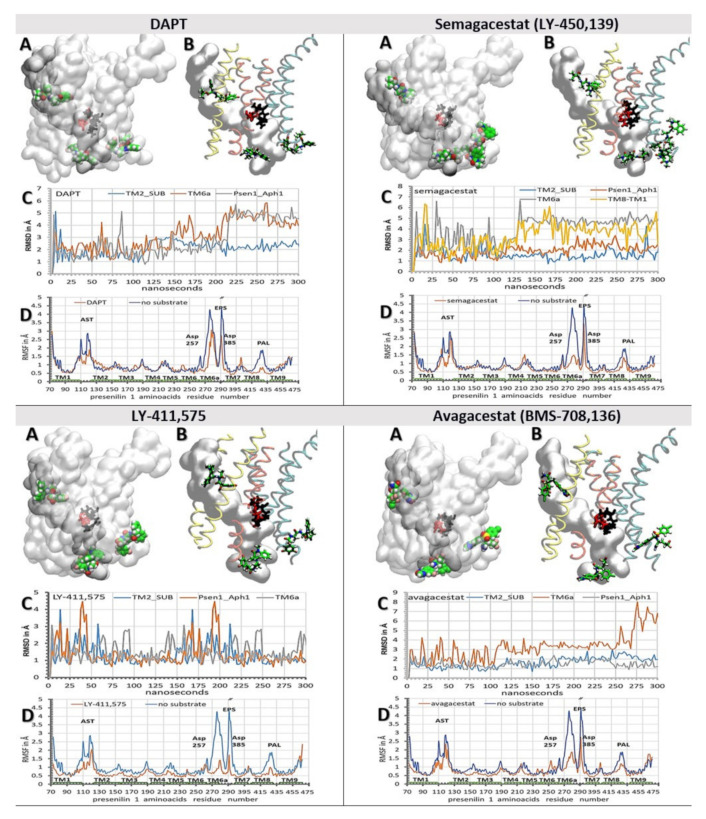
(**A**–**D**) Binding interactions between the biphasic drugs and γ-secretase in a complex with its substrate (PDB: 6IYC) [38]. The all-atom molecular dynamic studies [51] started with γ-secretase in a complex with different biphasic drugs from the molecular docking calculations (Figure 4) [48]. (**A**,**B**) Different drugs can bind to different sites next to the substrate. The presenilin structures are shown as transparent surfaces. Buried under the surface in the active site tunnel are the substrate (gray surfaces), the active site Asp 257 and Asp 385 (red licorice), and the PAL motif (black licorice, Pro 435, Ala 434, Leu 433). The drug molecules are shown as: carbon, green; oxygen, red; nitrogen, blue; fluorine and chlorine, pink, in order to make the structures underneath the surface visible. (**B**) TM2 and TM3 are shown in yellow; TM6, TM6a, and TM7 are shown in orange; TM8, TM9, and TM1 are shown in cyan. (**C**) RMSD values for the drugs bound at different sites are shown as a function of the simulated molecular time [51]. The steep increases in the RMSD values indicate the sliding of the drugs in the binding sites, while the fluctuations in the RMSD values indicate the relative mobility of the drugs in their binding sites. (**D**) The RMSF values for the individual amino acids show how the drugs can decrease the protein mobility at different sites [40,41]. Different structural elements are mapped on the graph: TM1-TM9, Trans-Membrane helix 1 to 9; AST, membrane-embedded opening of the Active Site Tunnel; EPS, Endo-Proteolytic Site; PAL motif (Pro 435, Ala 434, Leu 433) [42], and Asp 257 and Asp 385 in the active site.

**Figure 6 pharmaceutics-13-00514-f006:**
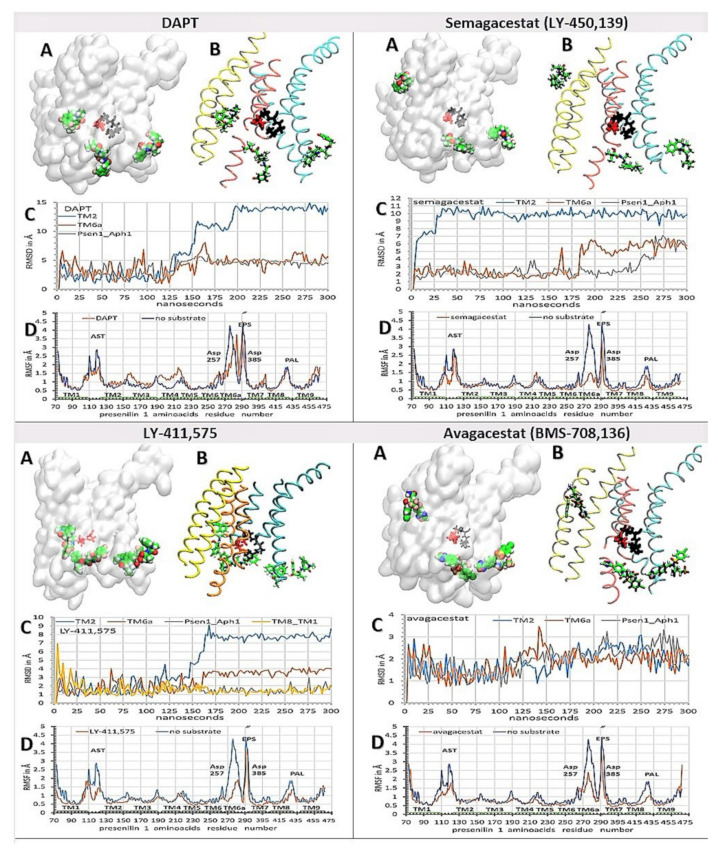
(**A**–**D**) Binding interactions between the biphasic drugs and γ-secretase in the absence of a substrate [38,39]. The all-atom molecular dynamics studies [51] started with γ-secretase in a complex with different biphasic drugs from the molecular docking calculations (Figure 4) [48]. (**A**,**B**) Different drugs can penetrate into the active site tunnel to a different extent. The presenilin structures are shown as transparent surface (**A**). TM2 and TM3 are shown in yellow; TM6, TM6a, and TM7 are shown in orange; TM8, TM9, and TM1 are shown in cyan (**B**). Buried under the presenilin surface are active site Asp 257 and Asp 384 (red licorice) and the PAL motif (Pro 435, Ala 434, Leu 433, black licorice) [42]. The drug molecules are shown as: carbon, green; oxygen, red; nitrogen, blue; fluorine and chlorine, pink. (**C**) The RMSD values for drugs bound at different sites are shown as a function of the simulated molecular time [51]. The steep increases in the RMSD values indicate the sliding of the drugs in their binding sites, while the fluctuations in the RMSD values indicate the relative mobility of the drugs in their binding sites. (**D**) The average RMSF values for the individual amino acids show changes in the protein mobility [40,41]. All four drugs can decrease the mobility at the specific sites in the presenilin structure to a different extent. Different structural elements are mapped on the graph. TM1-TM9, Trans-Membrane helix 1–9; AST, membrane-embedded opening of the Active Site Tunnel; EPS, Endo-Proteolytic Site; PAL, motif (a.a. 433 to 435) [42], and Asp 257 and Asp 385 in the active site.

**Figure 7 pharmaceutics-13-00514-f007:**
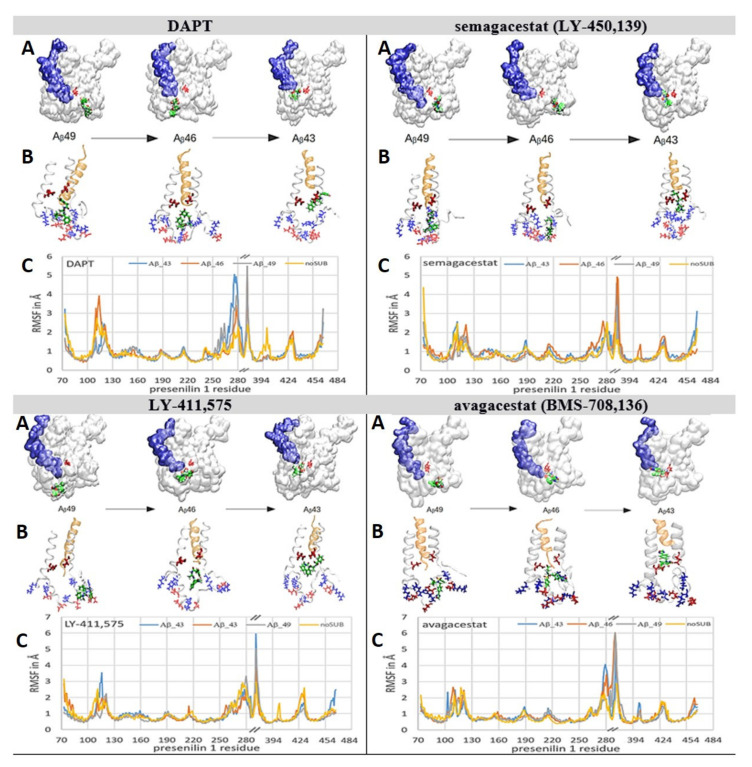
(**A**–**C**). Biphasic drugs can bind to γ-secretase and selectively interfere with the processive proteolytic cleavages. We used all-atom molecular dynamics studies [51] to analyze whether the molecules of biphasic drugs can penetrate into the active site tunnel when γ-secretase is in a complex with different Aβ catalytic intermediates. With all four drugs, the deepest penetration is observed with Aβ 43, and the lowest penetration is observed with Aβ 49. The penetrations are depicted using the presenilin structures [38], and quantitatively using RMSF values as a function of amino acid positions [40,41]. (**A**) The presenilin structures are shown as a white transparent surface in order to make the structures below the surface visible. Buried underneath the surface in the active site tunnel are different Aβ catalytic intermediates (blue surface) and the active sites Asp 257 and Asp 385 (red licorice). The drugs are shown as green Van der Waals models. (**B**) The white ribbon models depict amino acids 240 to 394 in presenilin structures. The gold ribbon models depict different Aβ catalytic intermediates. The positively-charged amino acids are shown as blue licorice; the negatively-charged amino acids are shown as red licorice, including the active sites Asp 257 and Asp 385. The drugs are depicted as green licorice models. (**C**) The RMSF values as a function of the amino acid positions show how biphasic drugs can affect different structural parts to a different extent with different Aβ catalytic intermediates [40,41].

## Data Availability

Data is contained within the article or Appendix A. The data presented in this study are available upon request at www.svedruziclab.com accessed on 2 April 2021.

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
