# Peer review of "Structural Analysis of the Simultaneous Activation and Inhibition of γ-Secretase Activity in the Development of Drugs for Alzheimer’s Disease"

_pharmaceutics, 2021, doi:10.3390/pharmaceutics13040514_

Round 1

Reviewer 1 Report

clearly written and well presented

Author Response

We are grateful  the reviewer for his time, efforts and supportive comments.

Reviewer 2 Report

Explaining what is the meaning and relevance of “biphasic response” will add clarity to the overall story.

"The great majority of drug-screening and optimization studies used unphysiological high saturation of γ-secretase with its substrate, and a limited concentration range for the drugs. Such an approach can be a good time and money-saving strategy in the early screening process [17–19]" : This information needs to be cited with most relevant and correct references.

Please explain the relevance of including information on avagacestat and semagacestat in the context of this study? Does the study aim to understand/investigate mechanisms underlying drug-induced cognitive decline? Please clarify.

Line 93-95 in Page 2 and 97-99 in page 3 seem repetitive, with same meaning.

One of the major gaps in this study is that it lacks the fundamental use of a negative control, a biphasic drug of similar biochemical/pharmacological features that doesnot interfere with gamma-secretase, and the use of vehicle control (DMSO in which the drugs were reconstituted).

The cell line name is inconsistent. Please check and make sure the name is correct as per guidelines.

What is the "n" for the data points in Figure 1?

Experiments conducted with all the 4 drugs in the study exhibit similarities in their biophysical profile, yet, leading to DAPT as weak and LY-411,575 as the strongest. What could be the reason underlying this difference? Any insights into the structure of these drugs?

What is the relevance of the findings from this study on the half-life of the drug-target interactions and how relevant is this interaction for target antagonism?

How does the findings of this study address the "cognitive decline" aspect of the biphasic drugs semagacestat and avagacestat? Addressing in the discussion will be very helpful and novel!

How this study will bridge the existing knowledge gap or in other words, the therapeutic impact of this study needs to be explained- Novelty is bit missing or is unclear in the current version. 

Author Response

Dear Reviewer 2

Dear Editors

We are grateful to the reviewer for her-his comments and extensive review of the manuscript. The presented comments have helped us to improve the text of the manuscript. Please find the individual responses embedded in the text of the review. 

R1. Explaining what is the meaning and relevance of “biphasic response” will add clarity to the overall story.

Answer:

The biphasic dose-response is described in very fine detail in Fig 1, in eqn.1 and in our earlier studies [1,2]. This is a very rare phenomenon that can be observed only with large and complex enzymes such as gamma-secretase. We are among world-leaders for mechanistic analysis of such complex phenomena (our biphasic studies have received about 50 citations in the last 7 years, what is a lot of citations for studies of enzymatic mechanism at the level of atoms)

We have several paragraphs that can explain why the studies of biphasic response are important. In the introduction section we have: “The mechanistic studies of the biphasic dose-response drugs are also important because these compounds can produce the same type of changes in γ-secretase activity as the disease-causing FAD mutations [1,3]. FAD mutation and the biphasic-drugs can cause accumulation of the longer and more hydrophobic Aβ products [4-8]. At low physiological substrate concentrations, the biphasic-drugs can increase the saturation of γ-secretase with its substrate like the Swedish mutation in its APP substrate [2]. At high substrate concentrations, the biphasic-drugs act as uncompetitive inhibitors, and show a decrease in the maximal turnover rates like the FAD mutations in presenilin 1 [1,2,4,5]. In essence, FAD mutations and biphasic-drugs have the same effect, a decrease in γ-secretase capacity to process its substrates [1]. A decrease in the catalytic capacity of γ-secretase can produce pathogenic changes in Aβ products with the wild type γ-secretase [1,5,9,10]. Disease-causing-mutations can affect activation and inhibition phases by different drugs to different extent [1,5]. Clinical studies showed that dose-dependent cognitive decline was observed in clinical trials with biphasic-drug avagacestat [11,12] and semagacestat [6,13-16].”

Several paragraphs in the discussion section has described the biomedical and pharmacological significance of the biphasic dose-response.

R2. "The great majority of drug-screening and optimization studies used unphysiological high saturation of γ-secretase with its substrate, and a limited concentration range for the drugs. Such an approach can be a good time and money-saving strategy in the early screening process [17–19]" : This information needs to be cited with the most relevant and correct references. 

              Answer:

Corrected. Please notice that we did not want to cite studies that did inadequate measurements in the past, since we did not feel that we want to hurt some people. Nevertheless, in the last paragraph in the discussion section, we have clearly stated that very expensive and toxic semagacestat disaster is caused by sloppy and inadequate measurements. 

R3. Please explain the relevance of including information on avagacestat and semagacestat in the context of this study? Does the study aim to understand/investigate mechanisms underlying drug-induced cognitive decline? Please clarify. 

Answer:

Yes, the reviewer has correctly identified our goals. We are delighted that the reviewer did not take our study as just a computational study, but as a study that can contribute to the understanding of Alzheimer’s disease and the development of the drugs. Both, semagacestat and avagacestat, were shown to have biphasic dose response in humans and model animals [6,13-16].. In the introduction section we wrote: 

Clinical studies showed that dose-dependent cognitive decline was observed in clinical trials with biphasic-drug avagacestat [11,12] and semagacestat [6,13-16]

At the end of the discussion section we wrote: 

The failed clinical trials with semagacestat [6,13,15-17] were much more expensive than the failed clinical trials with avagacestat [11,12]. In large part that can be attributed to better documented and more comprehensive preclinical studies with avagacestat [11,12,14,18,19].

Seamagacestat and avagacestat are biphasic dose-response drugs, and we showed in the past that biphasic dose response can produce the same changes in gamma-secretase activity as disease causing mutations [1,2,5]. In this study we show that semagacestat and avagacestat target the proteins sites that are affected by the diseases causing mutations. This gives a clear presentation of structure-activity relationship.

R4. Line 93-95 in Page 2 and 97-99 in page 3 seem repetitive, with same meaning. 

Answer:

Corrected.

Actually, the repetitive text has been repeated in 6 different places, in a different context, in the entire manuscript. The text is:”Clinical studies showed that dose-dependent cognitive decline was observed in clinical trials with biphasic-drug avagacestat [10,28] and semagacestat [7,16,29–31].” We have repeated that text many times, since the ability of the drugs to facilitate the progress of the disease has numerous implications, and frankly speaking is amazing. We have dedicated our current career, and all our funding, to explore similarities between the disease-causing-mutations and the drug activities at the molecular level. The reviewer would probably agree with us that cognitive decline induced with drugs is such an amazing observation that it deserves to be described from different aspects. Line 93-95 shows that statement in the light of shared enzymatic features between the drugs and disease causing mutants. Lines 97-99 have presented the aims of this study. Two related, but different issues.

R5. One of the major gaps in this study is that it lacks the fundamental use of negative control, a biphasic drug of similar biochemical/pharmacological features that does not interfere with gamma-secretase, and the use of vehicle control (DMSO in which the drugs were reconstituted).

Answer:

We apologize, there is a misunderstanding here. 

  1. We did the measurements with drug concentrations 0. We have specifically outlined in the text, that measurements of activity as zero concertation are crucial. It is a control measurement that can show that all cells have the same basic Abeta metabolism. The activities at zero drug concertation are value is crucial for the accurate interpretation of the biphasic inhibition as indicated at the start of the results section. 
  2. The computational studies have several embedded controls. In the last 10 years we have used enzyme-based [5], cell-based [1,2], and in_silico based studies (this study), to analyze and check the same thing: molecular mechanism behind biphasic dose-response. Thus our conclusions were evaluated and have been established on evidences that come from very different experiments and studies.
  3. In this study we have combined a number of different computational techniques, that all give a consistent message. We have combined coarse-grained studies (Fig 3), molecular docking studies (Fig 4), and all-atom molecular dynamics studies (Figs 5 to 7), to show that drugs bind to the most mobile parts of gamma-secretase structure. Chemoinformatics approaches are used to show consistency between the physical properties of the drugs and the presented binding sites and binding interactions (Figure 2). In total this manuscript has 7 figures in the main text, and in supplement 5 figures and a video. That is certainly more results than an average analysis of molecular mechanisms.
  4. Finally, the manuscript has over 70 references! We have related out results to biphasic dose-response studies in cells, model animals, and clinical trials 

R6. The cell line name is inconsistent. Please check and make sure the name is correct as per guidelines.

Corrected. 

R7. What is the "n" for the data points in Figure 1?

Answer:

Please notice that we are not doing simple measurements of Abeta 40 secretion from the cells. We are doing a much more complex study: a functional analysis of Abeta secretion from cells. We are looking at how Abeta secretion can fit to a model mechanism that represents drug binding to two different sites (Methods Eqn 1). The activation site and the inhibition site, with different Hills coefficients. The corresponding equation has six fitting parameters (Fig 1), out of those the initial activity and the final activity can be measured independently, as indicated in the text. EC50, IC50, and Hills coefficients can be calculated only if there are enough independent points, that are calculated as the number of degrees of freedom https://en.wikipedia.org/wiki/Degrees_of_freedom_(statistics) . We have 4 free fitting parameters and at least 10 data points for each drug, which indicates that we minimally have 6 degrees of freedom (i.e. 10-4). 

Please notice that data points are within 5% of the values calculated by the model curve. That is not surprising, since ELISA measures of Abeta 40 are so established and perfected, that the measured values have less than 10% error [1,2]. For that reason, we did not do error-bars, Abeta 40 measurements are so precise that the bars would make the graphs such a large number of data points overcrowded, with no significant new information. 

R8. Experiments conducted with all the 4 drugs in the study exhibit similarities in their biophysical profile, yet, leading to DAPT as weak and LY-411,575 as the strongest. What could be the reason underlying this difference? Any insights into the structure of these drugs?

Answer:

Yes, there are lots of structural insights. As indicated in the text and video 1 and Figure 3 in the supplemental material, DAPT is relatively narrow and flexible molecule, that can move around driven by the flexible protein loops (Figure 2). LY-411,575, is a much wider and less flexible molecule that can jam in the cavities on the protein surface and immobilize the flexible protein loops. Please notice that we say in the text that semagacestat has half of DAPT and LY411-575 features and shows intermediate binding constants. Avagacestat shares no structural similarities with LY-411,575, but shows the same effect, and the same binding affinity, because it is also a bulky molecule that can jam the flexible loops at each end of the active site channel.

R9. What is the relevance of the findings from this study on the half-life of the drug-target interactions and how relevant is this interaction for target antagonism?

Answer:

PK-PD processes and clinical pharmacology are not our expertise beyond the results presented in Figure 2, and the studies by Tong [12]. Data presented in Figure 2 shows more pharmacological analysis than many other biochemical studies that have analyzed drugs that target gamma-secretase. Some analysis of clinical pharmacology data is also shown in our earlier studies [1], which showed that the clinical data are fully consistent with our molecular mechanism. 

R10. How does the findings of this study address the "cognitive decline" aspect of the biphasic drugs semagacestat and avagacestat? Addressing in the discussion will be very helpful and novel!

Answer:

Please notice that we wrote in several places in the manuscript, that disease-causing mutations and drugs can produce the same type of changes in the gamma-secretase activity. In the last 10 years, we showed in different type of studies that the mutations and drugs depend on the same molecular mechanism [1,2,5,19]. Several paragraphs in discussion section and introduction section have described over 10 different studies that can link disease-causing mutations and the drugs. We have described the connection between our earlier studies and the studies by other labs in the last 20 years. In this study, we have shown using enzyme structures that the drug bind to sites targeted by the mutations, and to the proximal sites. Such binding can affect the mobility of the flexile protein loops that control the processive catalysis in the production of Abeta peptides of different lengths. We highlight that both the drugs and the mutations can affect dynamic protein loops that can affect the processive catalysis by gamma-secretase. Some outlines from the texts of the manuscript: 

R11. How this study will bridge the existing knowledge gap or in other words, the therapeutic impact of this study needs to be explained- Novelty is bit missing or is unclear in the current version. 

This is a little bit surprising and a harsh comment!?!?! Most of the experts on professional meeting that we have attended, told us that our results are so novel and provocative, that they want to see lots of additional results before they can accept the conclusions. We will sum some of the features that makes our work unique and important, we have indicated this points in the text:

  1. This work is certainly therapeutically significant since biphasic dose-responses are observed in clinical trials on humans [12], and in preclinical trials with model animals [19]. There will be NO development of drugs for Alzheimer’s disease if we do not have full understanding of this biphasic dose response phenomena. We are the only laboratory in the world that has described molecular mechanism behind this biphasic dose response down to the atomic details. We would be very grateful to the reviewer, if she-he can point to the other studies that have described the molecular mechanism behind this biphasic phenomena. We are the only lab in the world that can do at one site: enzyme-based, cell-based, and inslico-based studies of gamma-secretase and the related drug development efforts [1,2,5]. The group leader and the lead author in this study got training in biochemistry and cell biology of Alzheimer’s disease working 3 years on an Eli_Lily funded project in the laboratory of Bart de Strooper, one of the top three scientists in the world in studies of Alzheimer’s disease https://scholar.google.com/citations?user=3jx9ldcAAAAJ&hl=fr. Our current lab is funded and given full access to one of the 50 most powerful supercomputers at European universities. Several groups have published computational studies of gamma-secretase, no other group has used so many different and resource-demanding computation techniques to present their conclusions. We used up to 20 microseconds of coarse-grained molecular dynamics studies, and up to 600 nanoseconds of all-atom molecular dynamics studies. All-atom molecular dynamics studies had a membrane composed of 7 different lipids and in all cases more than 300 thousand atoms. Such calculations can take up to five days on up to 2400 microprocessors.

  1.  we are the first lab ever to point-out that the drugs and the disease-causing mutations can do the same changes in gamma-secretase activity. We would be very grateful to the reviewer if she-he can point to the other studies, that have presented arguments that drugs and the disease-causing mutations can do the same changes in gamma-secretase activity! We have shown the mutations-drug connections using enzyme-based, cell-based and now in_silico-based studies [1,2,5]. Three different techniques. We also highlight the strength of our conclusions by citing (i.e. interpreting) supporting evidences from published studies in the past 20 years. We have highlighted in the discussion section how biomolecular scientists in the future can feed the drugs to cells and model animals, and thus, in a highly controlled manner trigger or stop molecular processes that lead to the development of the disease. Please notice that our results section is about 90% computational studies, while our discussion section is 90% biomedical aspects of the diseases. 

  1. maybe we could add some bragging phrases in the manuscript. However, in the last 20 years lots of drugs have failed, becasuse there were too many bragging studies in the past, too many self-proclaimed experts that had no training in enzymology. With no training in enzymology, people have done really stupid mistakes, such as comparing different substrates and drugs without knowledge about the differences in Km and Kcat values. The result of such a sloppy and stupid approach were spectacular failures such as semagacestat, and avagacestat. Several thousand people got toxic drugs in clinical trials, and over a hundred million dollars have been wasted, all because people did not take into account basic enzymology rules in studies of enzyme-drug interactions. Little modesty can certainly help at this moment.

  1. Svedružić Ž, M.; Popović, K.; Šendula-Jengić, V. Decrease in catalytic capacity of γ-secretase can facilitate pathogenesis in sporadic and Familial Alzheimer's disease. Mol Cell Neurosci 2015, 67, 55-65, doi:10.1016/j.mcn.2015.06.002.
  2. Svedružić, Z.M.; Popovic, K.; Sendula-Jengic, V. Modulators of gamma-secretase activity can facilitate the toxic side-effects and pathogenesis of Alzheimer's disease. PLoS One 2013, 8, e50759, doi:10.1371/journal.pone.0050759.
  3. Hochard, A.; Oumata, N.; Bettayeb, K.; Gloulou, O.; Fant, X.; Durieu, E.; Buron, N.; Porceddu, M.; Borgne-Sanchez, A.; Galons, H., et al. Aftins increase amyloid-beta42, lower amyloid-beta38, and do not alter amyloid-beta40 extracellular production in vitro: toward a chemical model of Alzheimer's disease? J Alzheimers Dis 2013, 35, 107-120, doi:10.3233/jad-121777.
  4. Chavez-Gutierrez, L.; Bammens, L.; Benilova, I.; Vandersteen, A.; Benurwar, M.; Borgers, M.; Lismont, S.; Zhou, L.; Van Cleynenbreugel, S.; Esselmann, H., et al. The mechanism of gamma-Secretase dysfunction in familial Alzheimer disease. EMBO J 2012, 31, 2261-2274, doi:10.1038/emboj.2012.79.
  5. Svedružić, Z.M.; Popovic, K.; Smoljan, I.; Sendula-Jengic, V. Modulation of gamma-Secretase Activity by Multiple Enzyme-Substrate Interactions: Implications in Pathogenesis of Alzheimer's Disease. PLoS One 2012, 7, e32293.
  6. Tagami, S.; Yanagida, K.; Kodama, T.S.; Takami, M.; Mizuta, N.; Oyama, H.; Nishitomi, K.; Chiu, Y.W.; Okamoto, T.; Ikeuchi, T., et al. Semagacestat Is a Pseudo-Inhibitor of γ-Secretase. Cell Rep 2017, 21, 259-273, doi:10.1016/j.celrep.2017.09.032.
  7. Yagishita, S.; Morishima-Kawashima, M.; Ishiura, S.; Ihara, Y. Abeta46 is processed to Abeta40 and Abeta43, but not to Abeta42, in the low density membrane domains. J Biol Chem 2008, 283, 733-738.
  8. Yagishita, S.; Morishima-Kawashima, M.; Tanimura, Y.; Ishiura, S.; Ihara, Y. DAPT-induced intracellular accumulations of longer amyloid beta-proteins: further implications for the mechanism of intramembrane cleavage by gamma-secretase. Biochemistry 2006, 45, 3952-3960.
  9. Kakuda, N.; Funamoto, S.; Yagishita, S.; Takami, M.; Osawa, S.; Dohmae, N.; Ihara, Y. Equimolar production of amyloid beta-protein and amyloid precursor protein intracellular domain from beta-carboxyl-terminal fragment by gamma-secretase. J Biol Chem 2006, 281, 14776-14786.
  10. Yin, Y.I.; Bassit, B.; Zhu, L.; Yang, X.; Wang, C.; Li, Y.M. {gamma}-Secretase Substrate Concentration Modulates the Abeta42/Abeta40 Ratio: implications for Alzheimer's disease. J Biol Chem 2007, 282, 23639-23644.
  11. Coric, V.; van Dyck, C.H.; Salloway, S.; Andreasen, N.; Brody, M.; Richter, R.W.; Soininen, H.; Thein, S.; Shiovitz, T.; Pilcher, G., et al. Safety and Tolerability of the gamma-Secretase Inhibitor Avagacestat in a Phase 2 Study of Mild to Moderate Alzheimer Disease. Arch Neurol 2012, 1-12.
  12. Tong, G.; Wang, J.S.; Sverdlov, O.; Huang, S.P.; Slemmon, R.; Croop, R.; Castaneda, L.; Gu, H.; Wong, O.; Li, H., et al. Multicenter, Randomized, Double-Blind, Placebo-Controlled, Single-Ascending Dose Study of the Oral gamma-Secretase Inhibitor BMS-708163 (Avagacestat): Tolerability Profile, Pharmacokinetic Parameters, and Pharmacodynamic Markers. Clin Ther 2012, 34, 654-667.
  13. Mitani, Y.; Yarimizu, J.; Saita, K.; Uchino, H.; Akashiba, H.; Shitaka, Y.; Ni, K.; Matsuoka, N. Differential effects between gamma-secretase inhibitors and modulators on cognitive function in amyloid precursor protein-transgenic and nontransgenic mice. J Neurosci 2012, 32, 2037-2050.
  14. Tamayev, R.; D'Adamio, L. Inhibition of gamma-secretase worsens memory deficits in a genetically congruous mouse model of Danish dementia. Mol Neurodegener 2012, 7, 19.
  15. Doody, R.S.; Raman, R.; Farlow, M.; Iwatsubo, T.; Vellas, B.; Joffe, S.; Kieburtz, K.; He, F.; Sun, X.; Thomas, R.G., et al. A phase 3 trial of semagacestat for treatment of Alzheimer's disease. N Engl J Med 2013, 369, 341-350, doi:10.1056/NEJMoa1210951.
  16. Henley, D.B.; May, P.C.; Dean, R.A.; Siemers, E.R. Development of semagacestat (LY450139), a functional gamma-secretase inhibitor, for the treatment of Alzheimer's disease. Expert Opin Pharmacother 2009, 10, 1657-1664.
  17. Schor, N.F. What the halted phase III gamma-secretase inhibitor trial may (or may not) be telling us. Ann Neurol 2011, 69, 237-239.
  18. Jämsä, A.; Belda, O.; Edlund, M.; Lindström, E. BACE-1 inhibition prevents the γ-secretase inhibitor evoked Aβ rise in human neuroblastoma SH-SY5Y cells. J Biomed Sci 2011, 18, 76, doi:10.1186/1423-0127-18-76.
  19. Burton, C.R.; Meredith, J.E.; Barten, D.M.; Goldstein, M.E.; Krause, C.M.; Kieras, C.J.; Sisk, L.; Iben, L.G.; Polson, C.; Thompson, M.W., et al. The amyloid-beta rise and gamma-secretase inhibitor potency depend on the level of substrate expression. J Biol Chem 2008, 283, 22992-23003.

Reviewer 3 Report

The authors combined in vitro studies of accumulation of Abeta peptides in human neuroblastoma cells treated with 4 distinct biphasic inhibitors of gamma-secretase and in silico studies (molecular modellng and dynamics) to determine kinetics and structure-activity features of the drugs’ effects. The combined results revealed the binding of multiple molecules of the inhibitors to different sites of the presenilin structure, with different kinetic parameters. The authors concluded that the gamma-secretase activation phase is consequent to binding of the drugs leading to facilitated opening of the active site tunnel and formation of the E:S complex whereas the uncompetitive inhibition phase is due to binding of the inhibitors next to mobile regions of the enzyme that regulate processive catalysis. These data are of relevance for future design of drugs that solely inhibit the gamma-secretase, avoiding the activation phase that, like mutations in the enzyme that increase formation of Abeta products, worsens the cognitive decline in Alzheimer’s disease. Although well written, the text would benefit of shortening as it is very repetitive. Particularly, figures 5 and 6 could be rearranged to have side-by-side panels comparing the binding of the same drug with the enzyme in the absence and in the presence of its substrate. This would facilitate visual comparison of the modeled structures and consequently will reduce the length of the texts describing each structure separately and then again, when they are compared. Highlights should also be shortened, as 3 out of 4 show more than 85 characters including spaces A few typos to correct: - page 4, line 137: concentration (not concertation) - page 7, line 224: salt (not slat)

Author Response

Dear Reviewer 3

Dear Editors

We are grateful to the reviewer for her-his comments and extensive review of the manuscript. The presented comments have helped us to improve the text of the manuscript. Please find the individual responses embedded in the text of the review. 

The authors combined in vitro studies of accumulation of Abeta peptides in human neuroblastoma cells treated with 4 distinct biphasic inhibitors of gamma-secretase and in silico studies (molecular modellng and dynamics) to determine kinetics and structure-activity features of the drugs’ effects. The combined results revealed the binding of multiple molecules of the inhibitors to different sites of the presenilin structure, with different kinetic parameters. The authors concluded that the gamma-secretase activation phase is consequent to binding of the drugs leading to facilitated opening of the active site tunnel and formation of the E:S complex whereas the uncompetitive inhibition phase is due to binding of the inhibitors next to mobile regions of the enzyme that regulate processive catalysis. These data are of relevance for future design of drugs that solely inhibit the gamma-secretase, avoiding the activation phase that, like mutations in the enzyme that increase formation of Abeta products, worsens the cognitive decline in Alzheimer’s disease.

We are humbled and grateful to the reviewer for these kind comments. We are delighted to see that the review has seen the major ideas presented in the manuscript.

Although well written, the text would benefit of shortening as it is very repetitive.

We absolutely agree that the manuscript is in some aspects repetitive. Not only that the manuscript is repetitive, but the manuscript with the supplement, shows much more result figures than the majority of the similar manuscripts. Almost every figure is a full page figure with multiple panels. The manuscript also shows more references than the majority of similar manuscripts. We could cut this manuscript to smaller manuscripts, and thus, have more publications on the resume. Please understand, there is a strong reason for presenting the manuscript in its current form.

The results from this manuscript have been presented on different professional meetings, to different professionals. Medical doctors, clinical pharmacologist, medicinal chemist, cell biologist, biochemist and the people who work with model animals. It was clear to us from those meetings, that one of the biggest problem, in the current research and drug development efforts, is a poor communication and understanding between different professionals.

To address the problem of poor communication between different experts, the manuscript is written in “several expert languages”, with desire to attract as many different professionals as possible. The figure 1 is cell biology, the figure 2 medicinal chemistry, while the figures 3 to 7 are biochemistry of membrane proteins. The results section show computational biochemistry, discussion and conclusions present biomedical significance of our research. The manuscript would be about 50% shorter if we have targeted only biochemistry expert, since large parts of the results sections could have been in the discussion section.

Please, could you allow us to have a manuscript that can unify different aspect of the disease? We will in the near future write a review manuscript to address the problem of communication between the disciplines. However the review manuscripts can present only ideas. The research in this manuscript should motivate different types of experts to copy presented approaches and experiments to exploit shared features between the disease causing mutations and the drugs that target gamma-secretase.

We all want have as many citations as possible, we have invested lots of efforts trying to prepare such a manuscript, please allow us to have that.

Particularly, figures 5 and 6 could be rearranged to have side-by-side panels comparing the binding of the same drug with the enzyme in the absence and in the presence of its substrate. This would facilitate visual comparison of the modeled structures and consequently will reduce the length of the texts describing each structure separately and then again, when they are compared.

Corrected. We are grateful to the reviewer for helping us to correct that. We combined description of Figures 5 and 6 in the results section, and we have cut the corresponding text by 40%. The text of the figure legends is shorter by about 20%. The figures 5 and 6 have 4 panels, each panel has 3 subpanels. Combing figures 5 and 6 in one figure will give a figure with 8 panels, and in total 24 subpanels! Figures with 24 subpanels simply do not fit to one page.

Highlights should also be shortened, as 3 out of 4 show more than 85 characters including spaces

Corrected.

A few typos to correct: - page 4, line 137: concentration (not concertation) - page 7, line 224: salt (not slat).

Corrected. We are very grateful to the reviewer for helping us to correct these embarrassing mistakes.
